

# A crab swarm at an ecological hotspot: patchiness and population density from AUV observations at a coastal, tropical seamount

Jesús Pineda[1], Walter Cho[2], Victoria Starczak[1], Annette F. Govindarajan[1], Héctor M. Guzman[3], Yogesh Girdhar[4], Rusty C. Holleman[4,5], James Churchill[6], Hanumant Singh[4] and David K. Ralston[4]

[1] Biology Department, Woods Hole Oceanographic Institution, Woods Hole, MA, United States of America
[2] Biology Department, Point Loma Nazarene University, San Diego, CA, United States of America
[3] Smithsonian Tropical Research Institute, Balboa Ancon, Panama
[4] Applied Ocean Physics and Engineering Department, Woods Hole Oceanographic Institution, Woods Hole, MA, United States of America
[5] San Francisco Estuary Institute, Richmond, CA, United States of America
[6] Physical Oceanography Department, Woods Hole Oceanographic Institution, Woods Hole, MA, United States of America

Corresponding author
Jesús Pineda, jpinedaa@hotmail.com, jpineda@whoi.edu

## ABSTRACT

A research cruise to Hannibal Bank, a seamount and an ecological hotspot in the coastal eastern tropical Pacific Ocean off Panama, explored the zonation, biodiversity, and the ecological processes that contribute to the seamount's elevated biomass. Here we describe the spatial structure of a benthic anomuran red crab population, using submarine video and autonomous underwater vehicle (AUV) photographs. High density aggregations and a swarm of red crabs were associated with a dense turbid layer 4–10 m above the bottom. The high density aggregations were constrained to 355–385 m water depth over the Northwest flank of the seamount, although the crabs also occurred at lower densities in shallower waters (∼280 m) and in another location of the seamount. The crab aggregations occurred in hypoxic water, with oxygen levels of 0.04 ml/l. Barcoding of Hannibal red crabs, and pelagic red crabs sampled in a mass stranding event in 2015 at a beach in San Diego, California, USA, revealed that the Panamanian and the Californian crabs are likely the same species, *Pleuroncodes planipes*, and these findings represent an extension of the southern endrange of this species. Measurements along a 1.6 km transect revealed three high density aggregations, with the highest density up to 78 crabs/m², and that the crabs were patchily distributed. Crab density peaked in the middle of the patch, a density structure similar to that of swarming insects.

## INTRODUCTION

Seamounts are distinct oceanic habitats found in all oceans (*Wessel, Sandwell & Kim, 2010*), yet key first-order ecological processes are not well understood (*Clark et al., 2010*).

Communities of benthic species on seamounts are regionally isolated, with elevated, shallow rocky habitat patches surrounded by deep sedimentary plains. These two environmental axes, type of substrate (hard vs. soft), and depth (gradients in food, light, and oxygen), create horizontal and vertical patterns in faunal zonation (*Pitcher et al., 2008*; *Thresher et al., 2014*). These patterns are likely determined regionally by circulation and larval dispersal, and vertically by physical factors and biological interactions. In the pelagic environment, the trapping and concentration of pelagic planktonic biomass around seamounts, due to hydrodynamic and behavioral processes, result in local increase of predators—such as fish and marine mammals (*Klimley, Richert & Jorgensen, 2005*; *Morato et al., 2008*; *Morato et al., 2010*). Thus, seamounts are ecological hotspots in the sense that many biological and physical processes combine to produce high benthic and pelagic biomass, and higher biodiversity. Seamounts are productive—their shallow summits have been fished for centuries and the biomass of zooplankton is unusually high, but debate remains over the mechanism of pelagic biomass enrichment. A commonly cited hypothesis is that zooplankton and fish productivity result from phytoplankton growth due to topographic induced upwelling of nutrients to euphotic waters, but the importance of this mechanism has been recently challenged (*Genin & Dower, 2007*). Seamounts harbor valuable yet slow-growing resources, such as reef-building corals (e.g., scleractinians), black corals (e.g., antipatharians), soft-corals (e.g., gorgonians), and fish, some of which can live over 100 years (e.g., orange roughy) (*Koslow, 1997*). These habitats, however, have been undersampled and understudied, with less than 1% of all seamounts explored (*Clark et al., 2010*). The occurrence of seamounts in open oceans beyond national jurisdiction, and advances in deep-sea fishing practices have resulted in severe anthropogenic pressure on seamount populations which, due to their life history characteristics, are among the least resilient populations in the marine environment (*Koslow, 1997*; *Schlacher et al., 2010*).

*Pleuroncodes planipes* Stimpson, 1860 (superfamily Galatheoidea, family Muninidae, *Ahyong et al., 2010*) adult crabs, also known as red crabs, tuna crabs, squat lobsters, and "langostilla", occur in pelagic waters and in deep continental shelf and continental slope benthic habitats. Larvae and small individuals (∼<2.6 cm standard carapace length) tend to dominate the pelagic fraction off western Baja California, with larger organisms occurring exclusively in the benthos (*Boyd, 1967*).

Large individuals reproduce, but observations of pelagic ovigerous females and their larvae in waters over bathyal and abyssal depths (∼2,000–3,500 m) suggests that a fraction of the pelagic population can reproduce as well (*Longhurst & Seibert, 1971*). *Pleuroncodes planipes* can be extremely abundant, with accounts of dense pelagic patches up to 7–10 km (*Gómez-Gutiérrez et al., 2000*). See also the casual account of a 16 km patch by B. Shimada, quoted in *Boyd (1967)*. Off Baja California *P. planipes* is the main prey of large pelagic predators such as yellowfin tuna and skipjack tuna (*Alverson, 1963*). *P. planipes* is well adapted to its pelagic lifestyle, where it can feed both on phytoplankton, by specialized filtration, and on small zooplankton (*Longhurst, Lorenzen & Thomas, 1967*). On benthic habitats, galatheoid crabs are deposit feeders and scavangers (*Nicol, 1932*; *Lovrich & Thiel, 2011*). Benthic *P. planipes* ingest particulate organic matter (detritus associated with

sediments), phytoplankton cells, and small crustaceans, foraminiferans and radiolarians (*Aurioles-Gamboa & Pérez-Flores, 1997*). When feeding on bottom sediments containing diatoms, detritus and small organisms, galatheoid crabs' "third maxillipeds ...act as brooms" (*Nicol, 1932*), which would disturb and resuspend fine sediment.

Most studies on *Pleuroncodes planipes* have been done in pelagic waters, and have provided little information on the benthic habitat. *Boyd (1967)* found that benthic *P. planipes* ranged from ~100 to 300 m water depth off western Baja California, with smaller individuals found in shallower bottoms, and population densities up to 11/m². These distributions correlated with oxygen minima waters, with oxygen levels below 0.5 ml/l. *Boyd (1967)* and *Robinson & Gómez-Gutiérrez (1998)* found that some benthic individuals tend to migrate from the bottom to the upper water column. The typical northern geographic range end of *P. planipes* is somewhere in Baja California. Intermittently, particularly during the El Niño phase of the El Niño Southern Oscillation (ENSO), its geographic range expands northward to California (*Longhurst, 1966*; *Smith, 1985*). The southernmost geographic end range of *P. planipes* appears to be somewhere in Costa Rica (*Wicksten, 2012*), where it is thought to overlap with the northern range of *Pleuroncodes monodon* (*Macpherson et al., 2010*; *Wehrtmann et al., 2010*; *Wicksten, 2012*). The center of abundance of pelagic *P. planipes* is in western Baja California (*Longhurst, 1968*; *Brinton, 1979*; *Gómez-Gutiérrez et al., 2000*). The distribution and abundance of benthic *P. planipes* is not well documented, particularly south of Baja California.

We present findings from a research cruise to Hannibal Bank, a coastal seamount in the Gulf of Chiriquí, Eastern Tropical Pacific coastal ocean off Panama (Fig. 1). This cruise explored the mechanisms that contribute to high densities of benthic and pelagic organisms in an ecological hotspot and examined the seamount biodiversity and the benthic community zonation along the depth gradient. Work included (a) submarine dives to collect, film and observe firsthand the benthic habitats, and onboard DNA extractions of collected benthic invertebrates, (b) autonomous underwater vehicle (AUV) transects to map population densities of abundant benthic fauna, and (c) hydrographic and velocity measurements over the seamount using a conductivity, temperature, depth (CTD) and oxygen profiler and a hull-mounted acoustic Doppler current profiler. Hannibal seamount and its shallow top, Hannibal Bank, are within the recently created Coiba National Park, a UNESCO World Heritage Site, off the Pacific coast of Panama. Hannibal Bank harbors abundant large fish sustaining artisanal fisheries, and is a destination for international sport fishermen. The flat-topped triangular-shaped seamount rises from 450 m to ~40 m occupying an area of 83 km² (Fig. 2). Proximate to the continental shelf edge, it is ~20 km west of Coiba Island, 60 km from the main coast, and centered at about 07°24′N, 82°3′W (*Cunningham, Guzman & Bates, 2013*). Hannibal seamount communities are likely influenced by several physical processes, including synoptic upwelling from December to late April (*D'Croz & O'Dea, 2007*), low aragonite saturation state (*Manzello et al., 2008*), low oxygen sub-thermocline waters, low salinity from runoff and precipitation (~3 m yearly precipitation), sharp thermal stratification, large internal tides, and a 4 m tidal range (*Dana, 1975*; *Pineda, Reyns & Starczak, 2009*; *Starczak et al., 2011*).

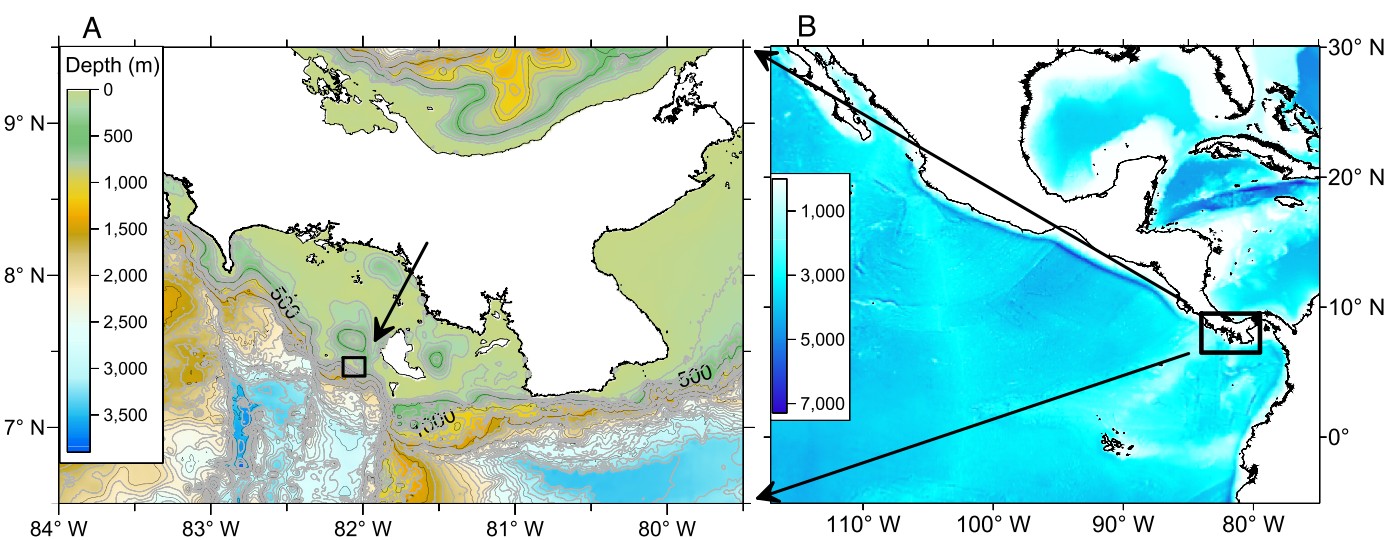

**Figure 1** **Map of the study area.** The box in (B) encloses the left panel, and the small box in (A) encloses Hannibal Seamount. Bathymetry data from GEBCO.

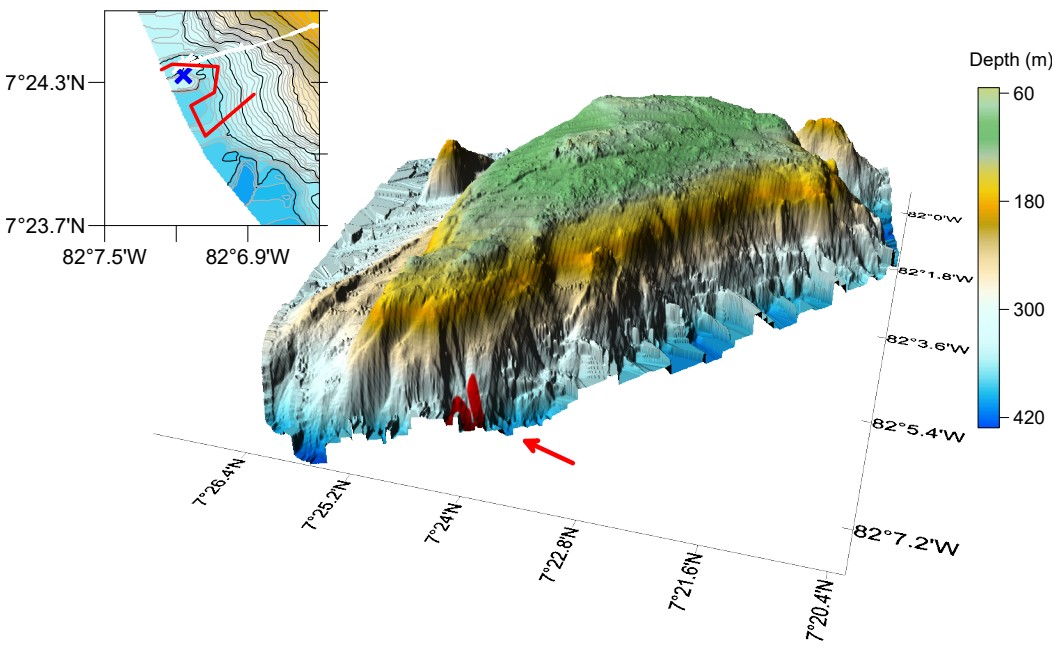

**Figure 2** **Hannibal Seamount, with location of the AUV transect indicated with a red line. Insert on the left delineates AUV transect, with end of transect near the blue cross.** The blue cross indicates the location of the CTD cast, and the submarine positions are in white. Depth data from *Cunningham, Guzman & Bates (2013)*.

On the last dive of the research cruise, we observed extraordinarily high densities of anomuran galatheoid crabs near the bottom of the seamount, and an associated turbid layer over the bottom. The encounter was unexpected and mesmerizing, and we documented these observations with high-definition video, a photo transect, environmental water

column measurements and genetic analysis of crab samples. Here we report on these observations, and address the following questions: What is the distribution of the crabs along a transect? What are the patterns of variability in abundance within a patch? Is there a relationship between the turbid layer and the crab aggregations? Are the crabs observed over the Hannibal Seamount the same species as *Pleuroncodes planipes* found off of California?

## METHODS

The cruise onboard the M/V *Alucia* from 31 March to 20 April 2015 focused on Hannibal Seamount. The "Ministerio del Ambiente de Panama" provided research and export permits, and the US State Department assisted in obtaining cruise permits. Work included ecological surveys over all flanks of the seamount (Fig. 2), and fifteen submarine dives conducted with Nadir, a 3-person submarine, and 11 dives with RV2, a 2-person submarine with more robust sampling capabilities than Nadir. On most missions, the submarines surveyed starting from the bottom of the seamount and continued to the top, working in tandem, within ~150 m of each other. Twelve transects with the Seabed autonomous underwater vehicle (AUV) complemented the diving missions and surveyed similar areas, collecting benthic imagery. On a typical cruise day, submarines were deployed during the morning and the Seabed AUV in the evening. Here we focus on observations completed on 18 and 19 April 2015, when crab aggregations were detected and studied. Further submarine and AUV observations on crabs were not possible due to technical issues and the cruise schedule. A conductivity, temperature, oxygen and depth profile was taken from the M/V *Alucia* using a Seabird SBE19 plus CTD in the vicinity of the submarine dive and Seabed AUV transect on 18 April 2015 (Fig. 2, blue cross in inset).

### AUV observations and density estimation

Seabed AUV conducted transects on the seamount, and obtained images to estimate densities of bottom organisms. Designed specifically for optical imaging of the seafloor (Singh et al., 2004b), the Seabed AUV has been used extensively for coral reef ecology, and other high resolution imaging applications (Singh et al., 2004a; Williams et al., 2014). It is equipped with high-dynamic range cameras (Singh et al., 2007) to provide species documentation via imagery that can be corrected for the nonlinear attenuation of light in the water. Seabed AUV navigated at a speed of ~20–25 cm/s and mean altitude of 3.5–4.5 m above the bottom along a predefined track, adjusting its altitude using a high frequency acoustic Doppler profiler. Seabed took 1,024 by 1,380 pixel images of the seafloor that was illuminated with a strobe, and recorded temperature, conductivity, depth, and altitude. The camera pixels are square and the field of view is 45° in the horizontal and 33° in the vertical. Image width, $x$, is determined from altitude (height above bottom), $z$, by noting that, $0.5(x/z) = Tan(45/2)$ which gives $x = 0.828z$. Because the pixels are square, the image height, $y$, is proportional to the number of pixels; i.e., $y = x(1024/1380)$, AUV specific altitude is used for every image, and image area is calculated as $x \cdot y$.

The Seabed AUV was programmed to take photographs every ~4 s, with image overlap. We examined every third image (12 s interval), which gave a sequence with no image

overlap. The non-analyzed images were used to resolve ambiguities in identification. Images were inspected for crabs and other organisms by eye, and all organisms were counted in each image.

Species identification of the crabs was confirmed by DNA barcoding of individuals in our samples (described below). Images from the Seabed AUV were light-corrected and inspected for organisms and type of substrate. A catalog of organisms was created from the photographs, and each morphospecies received a code. *Pleuroncodes planipes* were easily distinguished in the video recording taken from the submarine dives, and in the Seabed AUV images. To estimate crab density (#/m$^2$), the number of crabs was divided by estimated image area in each photograph.

## Patchiness estimate

Patchiness of *Pleuroncodes planipes* was estimated with $I_{mod}$ using the formula of *Bez (2000)* modified by *Décima, Ohman & De Robertis (2010)*. This index, based on Lloyd's index, considers a transect that does not sample the entire range of the species:

$$I_{mod} = \left[ \frac{\sum_i z_i^2}{s(\sum_i z_i)^2} \right] N$$

where $z_i$ is the density of the crabs in a given image, $s$ is the size of the sampling unit used in the survey (mean quadrat size, 8.93 m$^2$), and $N$ is the number of images analyzed. For comparison, we also report the patchiness index of an unidentified stomatopod that was easily detected in the photographs.

## Seamount sample collection and genetic barcoding

The submarines collected benthic organisms opportunistically, using a robotic manipulator arm, a net and a sediment scoop. We had a Ministerio del Ambiente de Panama permit # SE/A-18-15. Collected specimens were stored in a compartmentalized honeycomb quiver or in a larger "biobox". After the submarine dives, the sampled organisms were held in chilled seawater until they were photographed and labelled (e.g., Fig. 3B), preliminary taxonomic identification based on morphology was made, tissue was collected, and DNA extractions were performed onboard. Here, we focus on *Pleuroncodes* crabs. DNA was extracted using the DNEasy extraction kits (Qiagen) following the manufacturer's protocol. Upon return to the laboratory at Woods Hole, we conducted a genetic barcoding analysis on the crabs. A portion of the mitochondrial cytochrome c oxidase subunit I (COI) gene was amplified by PCR using the universal HCO-2198 and LCO-1490 primers (*Folmer et al., 1994*). PCR conditions were: 95 °C for 3 min; 35 cycles of 95 °C for 30 s, 48 °C for 30 s, and 72 °C for 1 min; and 72 °C for 5 min. PCR products were visualized on agarose gels stained with Sybr Safe (Life Technologies). PCR products were purified using Qiaquick PCR purification kits (Qiagen) and sequenced in both directions (MWG Eurofins Operon). Sequences were analyzed using the Geneious v. 7.1.7 software platform (Biomatters). Because morphological and video examination suggested that the crabs were *Pleuroncodes planipes*, we also sequenced COI from crabs identified as *P. planipes*
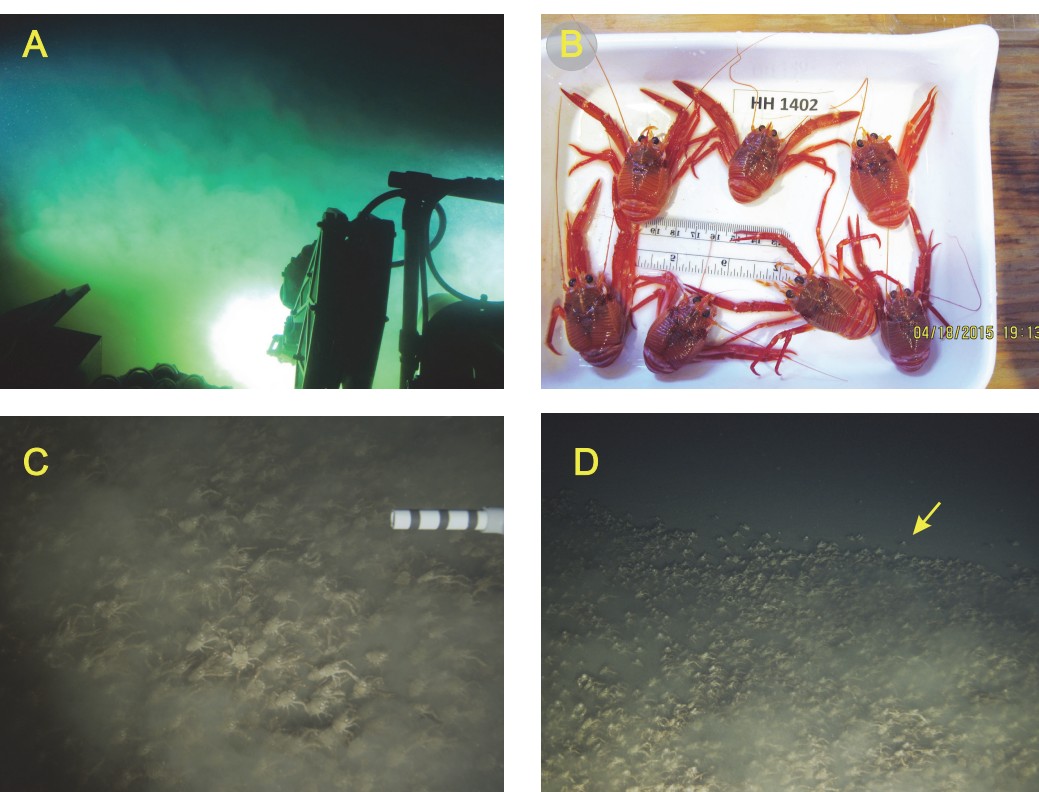

**Figure 3** **Photographs and video still frames of *Pleuroncodes planipes* and its environment.** (A) Image taken within Nadir as it approached the bottom, from about 6 m above the bottom, where *P. planipes* aggregations were first found. (B) *Pleuroncodes planipes* collected from the aggregation, with ruler scale in cm and English units. (C) Still frame from HD video of a dense patch of *P. planipes* on the bottom. The white PVC segment is about 20.5 cm long (D) Nearbed turbidity dropped at the edges of the *Pleuroncodes* patch. In the video the crabs were moving on the bottom towards the right, with a few crabs found beyond the boundary of the patch lagging behind the main aggregation. The crab marked with a yellow arrow was separate from the large patch and then merged into the patch by advancing in a direction perpendicular to the direction of patch movement.

from California for comparison (collection details below). Hannibal and California crab sequences were aligned with ClustalW (*Larkin et al., 2007*) using default parameters. The ends of the alignment were trimmed so that the dataset was complete for all taxa. Uncorrected $p$ and Kimura 2-parameter distances were calculated and a neighbor-joining tree was constructed in PAUP* (*Swofford, 2003*).

## Sample collection in a mass stranding event

From January to August 2015 there were several mass stranding events of *Pleuroncodes planipes* crabs on Southern California beaches, documented from news reports, the Lexis-Nexis database, and informal surveys (J Pineda, pers. obs., 2015, Table S1). In June 2015, crabs were observed in a San Diego beach (S Searcy, Univ. San Diego, pers. com., 2015, and J Pineda, pers. obs., 2015), and most of the crabs on the beach were still alive. At False Point, La Jolla (32°48′28.51″N, 117°15′54.96″), we collected galatheoid crabs on 2–5 June 2015, and preserved them in ethanol to provide reference specimens for DNA barcoding of seamount crabs.

## RESULTS

### Submarine observations and AUV mission

On the last diving mission of the cruise, 18 April 2015, the two submarines dived to the bottom by the Northwestern flank of the seamount (Fig. 2). Upon approaching the bottom, a very dense cloud of sediment was encountered; on no other submarine or AUV dive had such a dense cloud been observed (Fig. 3A). Altitude soundings from the submarine indicated that the turbid cloud extended 4–10 m over the ocean floor. As the submarine approached the bottom, a large number of galatheoid crabs were encountered. RV2 took 13 min and 40 s high-definition video of the crabs. A few still photographs and other video were taken from within the Nadir.

The video clips and photographs show that crabs were sometimes interacting among themselves (e.g., facing each other pulling out a dead crab) and with other organisms, including a sand eel. In some footage, crabs were sparsely distributed, and appeared to be sedentary. In other footage, benthic crabs were very dense, touching adjacent crabs, with most crabs moving broadly in the same direction (Figs. 3C and 3D) as a swarm (Video S1). In this footage, some crabs jumped and swam a few 10s of cm and landed in another spot. A crab outside of the patch moved towards, and merged with the main patch (Fig. 3D). Sand eel, small pelagic fish, shrimp, and a few stomatopods were in close proximity to the crab aggregation.

The population observed in the footage was composed of relatively large crabs, with no visible smaller individuals, i.e., $\sim$<2.3 cm carapace length. (See Fig. 3B for typical crabs, with $\sim$2.7 cm carapace length; carapace length as measured by *Gómez-Gutiérrez et al., 2000*). For most of the footage, the submarine hovered 2–3 m above the bottom, and the submarine and its lights did not appear to affect the behavior of the crabs. The high turbidity immediately above the bottom extended horizontally for at least 10s of m, and the turbid cloud appeared to be associated with the crab patch. As the two submarines moved up the seamount slope and abandoned the patch, the density of crabs decreased abruptly, and the turbid cloud disappeared (Fig. 3D).

On 19 April 2015, the Seabed AUV was programmed to complete a photo-transect in the same region as the crabs seen on 18 April. The AUV dived to about 325 m, and then completed a 1,610 m transect which included a set of turns to avoid potential high risk areas (e.g., rocky pinnacles) (Fig. 2, inset). Mean image width and length for this transect were 3.46 and 2.57 m, yielding a mean area per image of 8.93 m² ($n = 580$). Mean AUV altitude and speed was 4.18 m, and 0.23 m/s. The 580 analyzed photos were taken at 12 s intervals, and consecutive images had a gap of 2.78–2.57 = 0. 21 m. (See Fig. 4 for an image from the AUV, with the highest density of crabs detected in the transect, 77.2 individuals/m²).

At depths in the region where crab swarms were observed ($\sim$355–385 m, Northwest flank) and collected (see below), including areas with crabs and without crabs, substrate type was sedimentary, and sediments appeared to be fine sand and silt, with no rocky substrate.

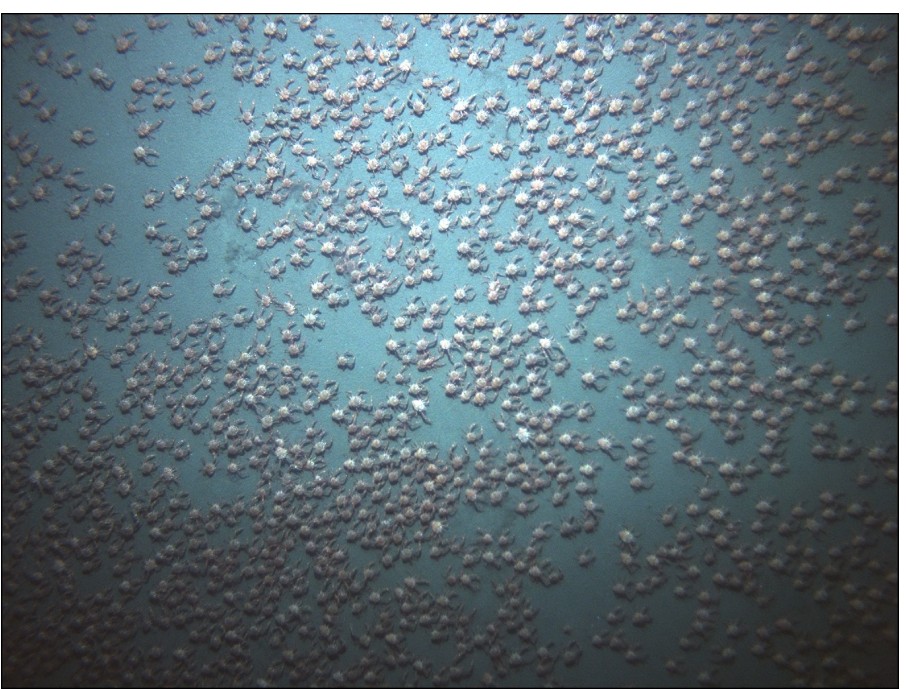

**Figure 4  AUV photograph with the highest density of *Pleuroncodes planipes*.**

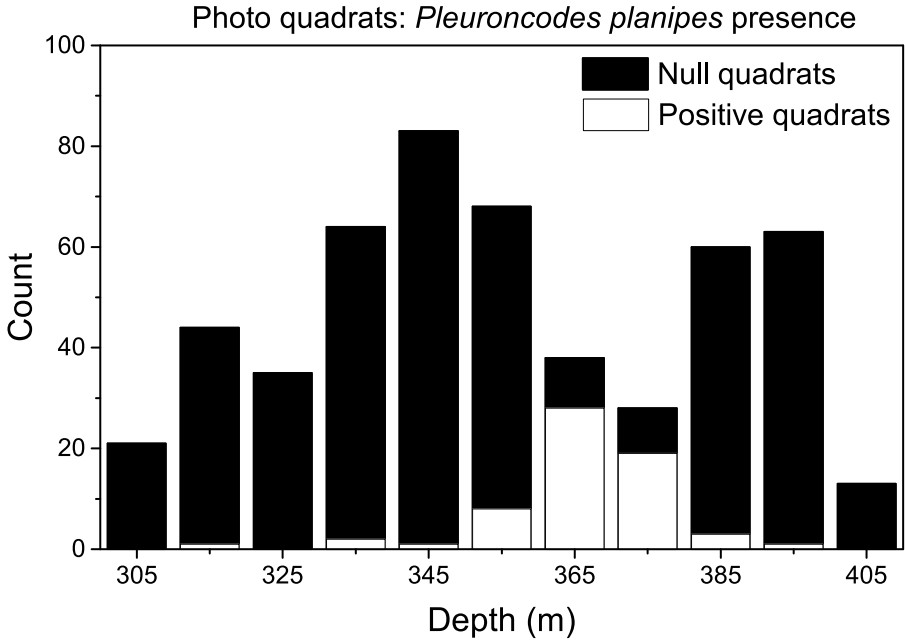

**Figure 5  Frequency distribution of quadrats with null and positive *Pleuroncodes planipes* counts.**  No
*P. planipes* occurred in null quadrats, whereas positive quadrats are those in which at least one *P. planipes*
was observed.

**Table 1  Pairwise distance comparisons for uncorrected *p* and K2P distance metrics.** Minimum and maximum pairwise distances (for all comparisons) and the mean distances for pairs within and between sampling localities are shown.

| | Minimum | Maximum | Within—Hannibal mean | Within—California mean | Hannibal—California mean |
|---|---|---|---|---|---|
| Uncorrected *p* | 0.00168 | 0.01513 | 0.01042 | 0.00336 | 0.00732 |
| K2P | 0.00168 | 0.01536 | 0.01055 | 0.00337 | 0.00734 |

### *Pleuroncodes planipes* abundance

*Pleuroncodes planipes* were detected in 12.2% of the Seabed AUV photographs. Images with counts of *P. planipes* tended to center around 365 m water depth (Fig. 5). Crabs were rare in the shallowest and deepest images, with bins centered at 305 and 405 m, although the number of images from these depths was low. Peak densities, with up to 72.2 crabs/m$^2$, occurred at 360–380 depths (Fig. 7). Three high-density patches were constrained to depths between 362 and 381 m (Fig. 8A), and were separated from each other by over 100's of meters along the northing (latitudinal) axis (Fig. 8B). The distribution of abundance in these peaks indicates that densities were low at the periphery, and that the maxima densities occur near the middle of the patch (Fig. 9). The distribution of crabs along the transect was very patchy, with $I_{mod} = 5.34$. Unidentified stomatopods that always occurred as singletons in the images had $I_{mod} = 3.54$. The turbid layer was not apparent in the Seabed AUV images.

### Galatheoid crabs DNA barcode ID

We obtained COI sequences for 6 specimens from Hannibal seamount and 4 specimens from the *Pleuroncodes planipes* stranding in California. Sequences were deposited in GenBank (Hannibal, KU179422, KU179423, KU179424, KU179425, KU179426, KU179431; La Jolla, KU179427, KU179428, KU179429, KU179430). Five out of the 6 Hannibal specimens were obtained from the main crab swarm on 18 April 2015. The 6th specimen was obtained on 3 April 2015, at a depth of 278 m, when crabs were observed on the bottom at the Northwest flank of the seamount (near 7°21.21′N, 82°1.37′W) at low densities. The final alignment for the combined seamount and California dataset was 595 base pairs. Inspection of the amino acid translation indicated that the sequences did not represent pseudogenes. Sequences differed from each other by between 1–8 base pairs. Uncorrected *p* and Kimura 2-parameter distances were similar to each other and ranged from 0.00168–0.01363. There were no shared haplotypes and the mean pairwise distance (for both metrics) between Hannibal specimens was greater than the mean distance between Hannibal and California specimens (Table 1 and Fig. 6).

### Water properties

The CTD cast revealed strong temperature, salinity, and oxygen stratification (Fig. 10). The temperature profile showed a sharp thermocline in the upper 40 m, with a temperature drop from 28.5 °C at the surface to 17.4 °C at 40 m, and a near-bottom temperature of 11.1 °C at ∼365 m. A halocline was also observed, with a salinity drop from 33.4 psu at the surface to 34.8 psu at 40 m. Maximum salinity occurred at mid depths (34.9 psu at 180 m), with a slight freshening with increasing depth (to 34.8 psu at 365 m).

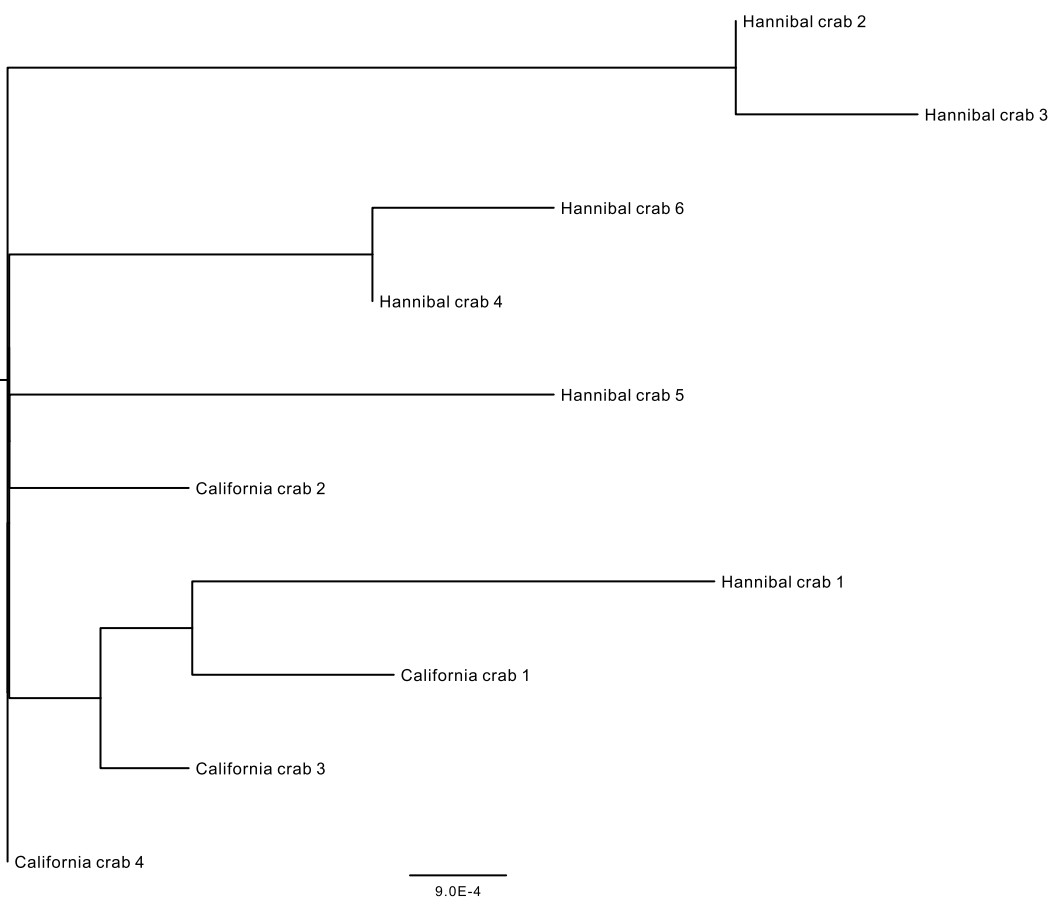

**Figure 6 Midpoint-rooted neighbor-joining topology based on mt COI Kimura 2-parameter distances.** Crab number 5 was found on 3 April at another location on Hannibal seamount, and was not in an aggregation.

Oxygen concentration decreased rapidly with depth, from over 4.8 ml/l at the surface to 1.1 ml/l at 50 m, and was less than 1.0 ml/l deeper than 250 m. The lowest oxygen value, 0.04 ml/l, was obtained from the deepest measurement, 365 m, ~15 m above the bottom. Thus, *Pleuroncodes planipes* maximum densities occurred at depths where waters were oxygen depleted. The vertical gradients of temperature and oxygen concentration changed abruptly at about 238 m, with larger gradients seen below 238 m. The vertical salinity also changed at around 238 m, but more subtly. Beam attenuation data from the SBE CTD revealed a turbid layer around 365 m depth in which optical attenuation tripled.

## DISCUSSION

Based on DNA barcoding, the Hannibal seamount crabs appear to be the same species as *Pleuroncodes planipes* from California. COI is the most typically used species barcode gene (*Bucklin, Steinke & Blanco-Bercial, 2011*), and sequence comparisons are frequently based on Kimura 2-parameter (K2P) distances (*Da Silva et al., 2011*). K2P distances may not necessarily be the best distance metric for a given taxon (*Srivathsan & Meier, 2012*; *Collins & Cruickshank, 2013*), other metrics may not necessarily perform better and the

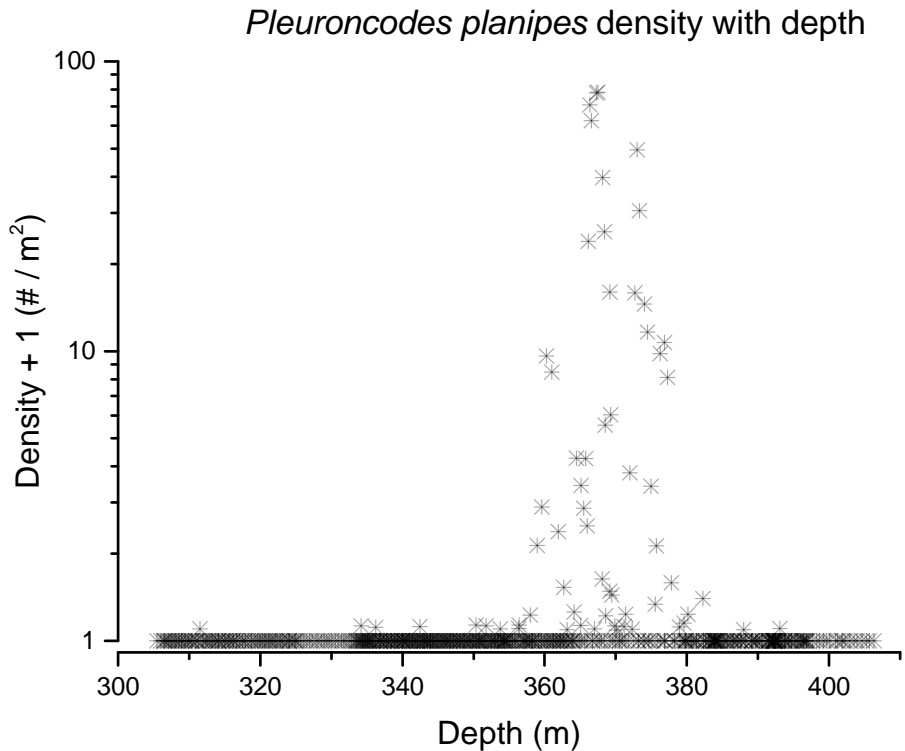

**Figure 7** ***Pleuroncodes planipes* density with depth.**

use of this metric permits straightforward comparisons with K2P distances from studies. Uncorrected *p* distances were similar to the K2P distances, and in both metrics, the mean distance between individuals at Hannibal Seamount was greater than the mean distance between Hannibal Seamount and California. Pairwise mitochondrial COI distances fell within the range of typical intraspecific distances for galatheoids (*Da Silva et al., 2011*). The southern range limit of *P. planipes* is considered poorly known (*Hendrickx & Harvey, 1999*), although researchers have suggested Costa Rica (*Wicksten, 2012*), and our observations here, supported by DNA sequences, may be the southernmost record.

Species have distinct patterns of variation in abundance over space, and understanding the factors that determine these patterns and their diversity is a central goal in ecology. Spatial distribution patterns may reflect individual and population processes, including settlement, dispersal, migration (*Roa & Tapia, 2000*) and behavior. For example, gregarious behavior and swarming in insects may produce characteristic spatial patterns of abundance (*Okubo & Chiang, 1974*). Whereas practically all organisms have patchy distributions at some spatial scale of observation, the causes and consequences of patchiness can reflect fundamental ecological and life history characteristics (*Marquet et al., 1993*). For example, patchiness can be species-specific and vary ontogenetically (*Hewitt, 1981*; *Décima, Ohman & De Robertis, 2010*), and species that face different degrees of patchiness may have evolved different life history strategies (e.g., *Dagg, 1977*). Patchiness, may be driven by external ("vectorial", environmental), reproductive, social (e.g., behavioral) and competitive ("coactive") processes (*Hutchinson, 1953*). Physical–biological interactions, such as the

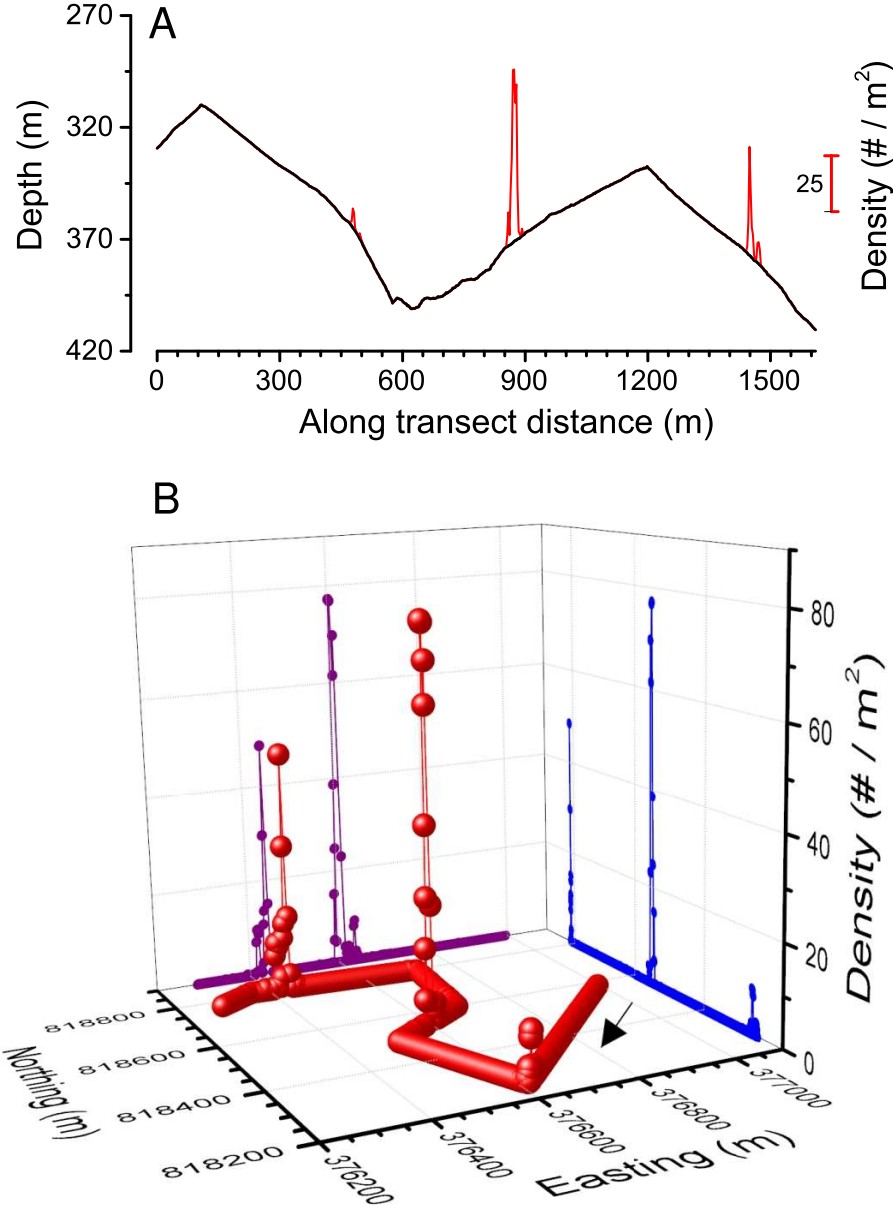

**Figure 8  Along transect *Pleuroncodes planipes* density on depth (A), and 3-d plot of density with latitude and longitude.** Arrow in (B) indicates the direction of the transect.

swimming up response of zooplankton and larvae to downwelling currents (*Scotti & Pineda, 2007*), might also produce patchiness (e.g., aggregation at fronts), and explain why only certain taxa aggregate in particular hydrodynamic settings.

The distribution of *Pleuroncodes planipes* was highly patchy, similar to other galatheoid populations (*Freire, González-Gurriarán & Olaso, 1992*; *Roa & Tapia, 2000*), and $I_{mod}$ values were higher than those of a stomatopod that occurred at smaller densities than *P. planipes*. The high *P. planipes* densities were constrained to a narrow subset of regions and depth ranges on Hannibal seamount. From the 26 submarine dives (15 missions to distinct

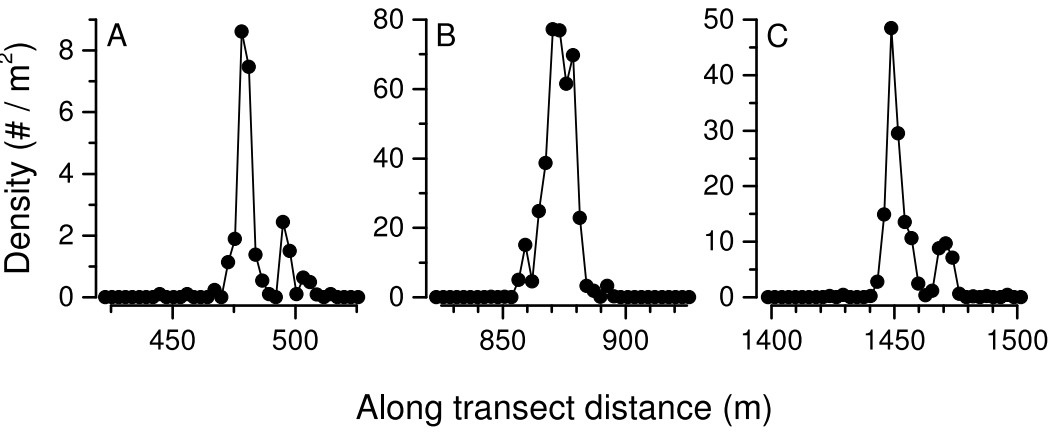

**Figure 9** *Pleuroncodes planipes* **abundance distribution in each of the three density peaks in Fig. 8.** For peak correspondence, see along transect distance and maximum density.

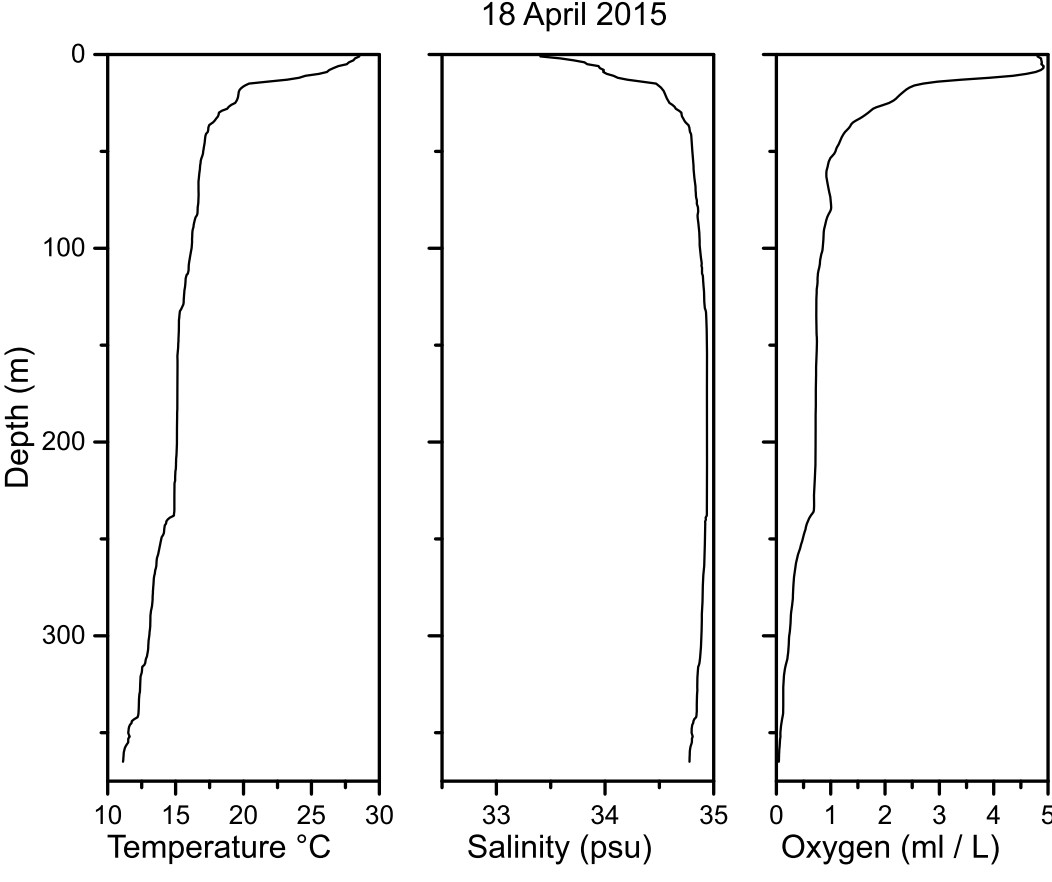

**Figure 10** **Temperature, salinity and oxygen profile measurements taken with a CTD on 18 April 2015 at a station a few tens of meters from the Seabed AUV transects.**

sites) and the 12 AUV transects, we observed dense aggregations of crabs in only one region, the Northwestern flank of the seamount, and these aggregations were constrained to ∼355–385 m water depths. The AUV survey detected three peaks in abundance (Figs. 9A–9C), and in peak B, the observed density was 77 individuals/m², one of the highest that have been measured for galatheoid crabs (*Lovrich & Thiel, 2011*, Table 6.1). Our sampling protocol cannot resolve whether these density peaks are discrete patches or whether aggregations were connected. It is unlikely, however, that crabs in density peak C were connected to crabs in peak B, because crab distributions were observed to be constrained to 355–385 m, and the crabs in B and C were separated by shallower depths (Figs. 2 and 8).

The density distribution within each of the three abundance peaks detected in the survey is consistent with a pattern where density increases towards the center of distribution (Fig. 9). However, we do not know whether the AUV surveyed the center of the patch. The two high-density peaks at ∼480 and 1,440 m along the transect (Figs. 9A and 9C) each have an adjoining lower-density peak. These lower-density peaks may represent budding, small aggregations that have split from the main aggregation, and might grow into larger patches, or they might merge into the larger, adjacent patch. These density distribution patterns are likely due to aggregation driven by the crab's gregarious behavior, and coordinated movement of the aggregation, a phenomenon that has been called swarming. *Okubo, Grünbaum & Edelstein-Keshet (2001)* describe swarming as a phenomenon where a group of organisms move together. Swarms are arguably one of the few ecological phenomena that possess emergent properties, where the characteristics of the aggregation cannot be simply explained by adding the individual's behaviors (*Parrish & Edelstein-Keshet, 1999*). In these complex systems, a focus on individual behavior is unlikely to explain the properties of the swarm. Whether all emergent properties in swarms are functional or not, is an open question (*Parrish & Edelstein-Keshet, 1999*). The increase in density towards the center is consistent with other organisms that form swarms and aggregations (e.g., insects, krill and schooling fish, *Okubo & Chiang, 1974*; *Okubo, Grünbaum & Edelstein-Keshet, 2001*), and patterns of abundance in other gregarious benthic populations where density increases towards the middle of the distribution might reveal a swarming behavior. Two other galatheoid species have patterns where density increases towards the middle of the patch (*Freire, González-Gurriarán & Olaso, 1992*) but in these European *Munida* spp. species, the scale of the patches is a few tens of kilometers, compared to the ∼100 m scale observed in our study. It is unclear whether the abundance structure of these *Munida* spp. and *Pleuroncodes planipes* patches are caused by the same processes. Dense benthic aggregations of other species of anomuran and brachyuran crabs (king crabs, spider crabs, tanner crabs, lyre crabs) have been observed, and some were related to reproduction (*Powell & Nickerson, 1965*; *DeGoursey & Auster, 1992*; *Stevens, Donaldson & Haaga, 1992*; *Stevens, Haaga & Donaldson, 1994*).

Crabs densities appeared to be higher and more clumped in the submarine video observations than in the AUV images (compare submarine video still frame Fig. 3 with AUV Fig. 4). The submarine video still frames in Fig. 3 were taken when *Pleuroncodes planipes* were moving as a group, a swarm, and most organisms appeared to be oriented in the same direction. In the AUV images, a consistent crab orientation and the swarm

motion were not obvious. Moreover, the turbid cloud observed from the submarine (Fig. 3) was not seen in any of the AUV images. The turbidity cloud was most likely produced by *P. planipes* activities, as the turbidity disappeared at the edge of the patch, and was not observed outside of the patch or in any other submarine dive or AUV missions. Diurnal patterns of activity might explain the differences in turbidity. Submarine observations were early in the day, whereas the AUV survey was done in the evening. However, another possibility is that crabs observed from the submarine were in a location with finer, and hence more easily suspended sediments than those surveyed by the AUV survey. However, the locations were not far from each other (Fig. 2).

The resuspension of sediment initiated by crab activity may affect the benthic environment over the Northwest seamount flank. Feeding of king crabs in waters ∼3 m deep off Kodiak Island, Alaska, resulted in a dense cloud of turbid water (*Stevens & Jewett, 2014*), and *Yahel et al. (2008)* found that bottom fish activity was an important mechanism for sediment resuspension and remineralization of organic matter between water depths of ∼60 and 140 m in Saanich Inlet (Vancouver Island, BC, Canada).

*Pleuroncodes planipes* occurred at water depths with very low oxygen (0.04 ml/l at ∼15 m above the bottom where the crabs were observed). The affinity of some galatheoids to low oxygen waters, and *P. planipes* in particular, is known (*Boyd, 1967*; *Lovrich & Thiel, 2011*). Depth distribution of *P. planipes* and other galatheoids might be related to these low oxygen levels (discussed by *Lovrich & Thiel, 2011*), but more study is needed to test this hypothesis.

*Pleuroncodes planipes* occurs in very high densities in the pelagic environment, and this species mass strands yearly in shallow water and intertidal beaches near the center of its pelagic abundance, Bahía Magdalena, Baja California (*Aurioles-Gaamboa, Castro-González & Pérez-Flores, 1994*), and more occasionally on California beaches (Table 1, *Longhurst, 1966*; *Smith, 1985*). While we were on hydrographic stations over Hannibal seamount and surrounding areas, we occasionally observed organisms that appeared to be pelagic red crabs swimming swiftly by the stern of the boat at night, illuminated by the vessel lights. Despite multiple attempts, we were not able to capture a specimen to assess its identity, so the occurrence of *P. planipes* in the water column above Hannibal seamount is unknown.

Our observations in Panama were conducted at roughly the same time when mass stranding events were registered in Southern California (Table S1), and the Hannibal and Californian individuals appear to be the same species based on their mtCOI sequences. Mass stranding of *Pleuroncodes planipes* in Southern California beaches has been linked to El Niño (*Smith, 1985*). A full El Niño had not been declared for January–June 2015, when many stranding events were reported (Table S1). On the other hand, an unusually large pool of warm water developed in late 2013 and early 2014 in the coastal temperate eastern Pacific, and persisted through much of 2015 (*Bond et al., 2015*), apparently unrelated to El Niño. The current forecast (February 2015, by NOAA Climate Prediction Center), indicate that the anomalous warm-water pool condition has been followed by an El Niño, and that a full El Niño is currently in progress. The "pool of warm water" conditions in January–June 2015 may be related to anomalously warm waters observed in Southern California's nearshore in fall 2014 (N Reyns, J Pineda & S Lentz, pers. obs., 2015). These conditions may help explain the appearance of *P. planipes* in Southern California, as speculated by

some news outlets. Whereas it is unlikely that our observations of benthic aggregations at Hannibal are connected with the California mass stranding events, it is significant that *P. planipes* can be simultaneously abundant at the two distant locations and at two different habitats. The high densities of *P. planipes* likely impacted local pelagic, intertidal, and deep seamount food webs.

Allochthonous supply of biomass, where resources from one habitat or ecosystem subsidizes another system, influences local population community and dynamics (*Polis, Anderson & Holt, 1997*). Moreover, the episodic availability of large quantities of biomass to benthic and pelagic organisms and marine mammals, including the supply of terrestrial material and whale carcasses to benthic deep sea communities, the mass stranding of pelagic organisms in shallow habitats, and the sudden availability of a new resource, represent an opportunistic yet important source of nutrition to the "receiving" communities (*Polis, Anderson & Holt, 1997*), and can influence food web structure and demographic rates (*Watt, Siniff & Estes, 2000*). The massive availability of *Pleuroncodes planipes* might influence diverse food webs.

Because of its pelagic and benthic lifestyle, and its abundance, *Pleuroncodes planipes* likely plays an important role in some seamount, continental shelf and shallow water food webs in the subtropical and subtemperate eastern Pacific. Several authors have noticed the key role of *P. planipes* in the pelagic environment, by virtue of its abundance and trophic role (*Alverson, 1963*; *Longhurst, 1966*; *Longhurst, Lorenzen & Thomas, 1967*; *Gómez-Gutiérrez et al., 2000*; *Robinson, Anislado & Lopez, 2004*). *P. planipes* was patchy but very abundant at Hannibal, and it might represent an important resource for pelagic predators at the seamount. More research is needed to assess the distribution and abundance of benthic *P. planipes*, as well as its potentially key role in subtropical and subtemperate eastern Pacific seamount and continental shelf habitats.

## ACKNOWLEDGEMENTS

We would like to express gratitude for the help and logistical support of the Captain and the crew of the M/V *Alucia*, the *Alucia*'s submarine team, Jeff Anderson, for programming and running the AUV operations, and keeping it safe, Alex Bocconcelli, WHOI's Marine Operations office support, and S Searcy (USD) for red crab stranding information in San Diego. Bathymetric data in Fig. 1 derived from the GEBCO_2014 Grid, www.gebco.net.

### Funding

This work was sponsored by a grant from the Dalio Foundation, Inc, through the Woods Hole Oceanographic Institution. The funders had no role in study design, data collection and analysis, decision to publish, or preparation of the manuscript.

### Grant Disclosures

The following grant information was disclosed by the authors:
Dalio Foundation, Inc.

## Competing Interests

The authors declare there are no competing interests.

## Author Contributions

- Jesús Pineda conceived and designed the experiments, performed the experiments, analyzed the data, contributed reagents/materials/analysis tools, wrote the paper, prepared figures and/or tables, reviewed drafts of the paper.
- Walter Cho and Annette F. Govindarajan conceived and designed the experiments, performed the experiments, analyzed the data, contributed reagents/materials/analysis tools, prepared figures and/or tables, reviewed drafts of the paper.
- Victoria Starczak, Héctor M. Guzman, Rusty C. Holleman and James Churchill conceived and designed the experiments, performed the experiments, reviewed drafts of the paper.
- Yogesh Girdhar conceived and designed the experiments, performed the experiments, contributed reagents/materials/analysis tools, reviewed drafts of the paper.
- Hanumant Singh contributed reagents/materials/analysis tools.
- David K. Ralston reviewed drafts of the paper.

## Field Study Permissions

The following information was supplied relating to field study approvals (i.e., approving body and any reference numbers):

The Ministerio del Ambiente de Panama provided permission (# SE/A-18-15).

## DNA Deposition

The following information was supplied regarding the deposition of DNA sequences:

GenBank (numbers also listed in the paper):

Pplanipes.sqn Hannibal_1402a KU179422

Pplanipes.sqn Hannibal_1402b KU179423

Pplanipes.sqn Hannibal_1402c KU179424

Pplanipes.sqn Hannibal_1402d KU179425

Pplanipes.sqn Hannibal_1402e KU179426

Pplanipes.sqn CAcrab1 KU179427

Pplanipes.sqn CAcrab2 KU179428

Pplanipes.sqn CAcrab3 KU179429

Pplanipes.sqn CAcrab4 KU179430

Pplanipes.sqn Hannibal_1051 KU179431.

## Data Availability

The raw data has been supplied as Data S1 and S2.

## Supplemental Information

Supplemental information for this article can be found online at http://dx.doi.org/10.7717/peerj.1770#supplemental-information.

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
