# Peer review of "A crab swarm at an ecological hotspot: patchiness and population density from AUV observations at a coastal, tropical seamount"

_PeerJ, doi:10.7717/peerj.1770_

## Round 0.1 · original submission · Minor Revisions

Dear Authors,

Thank you for submitting your manuscript to PeerJ. Please note that the reviewers have highlighted a few minor corrections you may wish to consider.

Best wishes,

·

Basic reporting

Article is clear and well written. The structure conforms to PeerJ standards.
The introduction includes relevant literature about seamounts, the study area and Pleuroncodes planipes behavior.
The figures are relevant and of high quality.
Raw data was supplied.

Experimental design

Research questions are not well defined and the sampling appears to be "opportunistic." (This was because many red crabs appeared on the last research dive.) However the results obtained are relevant and meaningful for the study area.
Methods are well described.

Validity of the findings

Research questions:
1-What is the distribution of the crabs along a transect?
Lines 248-249 “The distribution of crabs along the transect was very patchy, with
Imod = 5.34”. Along this transect the spatial distribution patterns was determined. However, sampling does not resolve how red crabs are distributed in the area and whether this distribution is related to environmental variables. For example, Is the patching distribution related to the type of substrate (rocky versus soft bottom)? Some information about substrate must be provided.
Why was the transect not replicated? I suppose this was due to the "opportunist" feature of sampling.

2-What are the patterns of variability in abundance within a patch?
This research question was answered in Lines 320-321 “The density distribution within each of the three abundance peaks detected in the survey is consistent with a pattern where density increases toward the center of distribution”.

3-Is there a relationship between the turbid layer and the crab aggregations? The relationship between aggregations of red crabs and turbidity is based only on observations from a submarine dive and video recording. Lines 227-230 “The high turbidity immediately above the bottom extended horizontally for at least 10’s of m, and the turbid cloud appeared to be associated with the crab patch. As the submarine moved up the seamount slope and abandoned the patch, the density of crabs decreased abruptly, and the turbid cloud disappeared”. This observation cannot be validated.
Some additional information's with statistical analysis of correlation between density or crab’s activity vs. presence/absence of turbid layer and other parameters (grain size) must be given.

Lines 353-355 “Moreover, the turbid cloud observed from the submarine (Fig. 3) was not seen in any of the AUV images. The turbidity cloud was most likely produced by Pleuroncodes planipes activities, as the turbidity disappeared at the edge of the patch, and was not observed outside of the patch or in any other submarine dive or AUV missions.”
Why did you not repeat the transect with the AUV in the evening to determine if the turbidity layer was related to the crab's activity patterns?

4-Are the crabs observed over the Hannibal Seamount the same species as Pleuroncodes planipes found off California?
Lines 280-281 “Based on DNA barcoding, the Hannibal seamount crabs appear to be the same species as Pleuroncodes planipes from California” DNA analysis confirm that it is the same species. It is the first study of this species in the area and provides interesting data on density and behavior, also it presents important results as the southernmost record of P. planipes.

Additional comments

I Recommend acceptance with minor revisions. Publication of this article is important for planning further studies that require knowledge of the abundance, behavior and trophic importance of P. planipes in this region.

·

Basic reporting

This work resulted from an expedition to Hannibal Bank; it presents interesting findings regarding population distribution of the crab Pleuroncodes planipes. It meets PeerJ standards, very well written and fully supported with supplementary material.

Experimental design

Within the restrictions that an observational study like this impose, this work rigorously analyze data and goes further to propose new hypothesis, with clearly stated research questions.

Validity of the findings

Statistical analysis are solid and validate their results. Their explanation about patchiness of this crab is consistent with current theory and their proposal of Pleuroncodes planipes is supported by genetic analysis.

Additional comments

It would be interesting to go further in the explanation of the turbidity layer as well as the abundance of this crab at so low OD.

·

Basic reporting

The article is written using clear and unambiguous text in English, includes sufficient introduction and background to demonstrate how the work fits into the broader field of knowledge.
The structure of the submitted article conforms to one of the templates
Figures are relevant to the content of the article, of sufficient resolution and appropriately described and labeled.
The submission is ‘self-contained,’ represent an appropriate ‘unit of publication’, and includes all results relevant to the hypothesis.

Experimental design

The submission describes original primary research within the Scope of the Journal, clearly defines the research question, and statements are made as to how the study contributes to answer it.
The investigation was conducted rigorously and to a high technical standardm and methods are described with sufficient information to be reproducible by another investigator.

Validity of the findings

The conclusions are appropriately stated, connected to the original question investigated, and limited to those supported by the results.

Additional comments

- Review In-text-citations, (use colon after Last names and et al.)(I did not correct all citations). List three authors in the citation, like in (Wessel, Sandwell & Kim, 2010 instead of Wessel et al. 2010) according to Preparing your submission instructions.

I have some taxonomic differences with the authors: according to Macpherson and Baba (2011) Pleuroncodes planipes is included in the family Munididae (Ahyong et al. 2010), and therefore should be called munidids and not galatheids. Besides, the authors use the term red crab for the red squat lobster P. planipes, I suggest making this change throughout the manuscript, because this better represents the real vernacular name of the species.
Ahyong ST, Baba K, Macpherson E, Poore, GCB (2010) A new classification of the Galatheoidea (Crustacea: Decapoda: Anomura). Zootaxa 2676, 57–68.
Macpherson E and Baba K (2011). Chapter 2. Taxonomy of squat lobsters. In: The biology of squat lobsters. Crustacean Issues Vol. 20. (Eds. GCB Poore, ST Ahyong and J Taylor) pp. 39–72 (CSIRO Publishing: Melbourne and CRC Press: Boca Raton).

---

## Round 0.2 · accepted · Accept

Dear Authors,

Thank you for submitting such an interesting manuscript to PeerJ and addressing those minor corrections. The supplementary video is quite extraordinary and a pleasure to watch.